# Evaluating structure selection in the hydrothermal growth of $FeS_2$ pyrite and marcasite

Daniil A. Kitchaev[1] & Gerbrand Ceder[1,2,3]

While the *ab initio* prediction of the properties of solids and their optimization towards new proposed materials is becoming established, little predictive theory exists as to which metastable materials can be made and how, impeding their experimental realization. Here we propose a quasi-thermodynamic framework for predicting the hydrothermal synthetic accessibility of metastable materials and apply this model to understanding the phase selection between the pyrite and marcasite polymorphs of $FeS_2$. We demonstrate that phase selection in this system can be explained by the surface stability of the two phases as a function of ambient pH within nano-size regimes relevant to nucleation. This result suggests that a first-principles understanding of nano-size phase stability in realistic synthesis environments can serve to explain or predict the synthetic accessibility of structural polymorphs, providing a guideline to experimental synthesis via efficient computational materials design.

[1] Department of Materials Science and Engineering, MIT, Cambridge, Massachusetts 02139, USA. [2] Department of Materials Science and Engineering, UC Berkeley, Berkeley, Calirfornia 94720, USA. [3] Materials Science Division, Lawrence Berkeley National Laboratory, Berkeley, California 94720, USA. Correspondence and requests for materials should be addressed to D.A.K. (email: dkitch@mit.edu) or to G.C. (email: gceder@berkeley.edu).

Nucleation and growth from solution remains one of the most experimentally and geologically important synthesis methods for crystalline solids. Hydrothermal growth, which involves precipitation from a superheated aqueous solution of precursor salts, is a particularly common route for natural mineral formation and synthetic single-crystal growth[1]. Despite the importance of this method, recipes for the hydrothermal growth of target solid phases remain largely empirical. The absence of predictive approaches to synthesis impedes the realization of novel compounds predicted by highly developed and high-throughput computational materials design approaches[2–5]. In addition, the computational screening of novel materials requires an efficient filter for synthesizability, which has in the past been restricted to the thermodynamic stability of the material of interest and empirical 'structural similarity' arguments[3,6].

We propose a general framework within which to view hydrothermal growth in a computationally accessible and systematic manner. We hypothesize that with a judicious choice of conditions, a synthesis can be designed such that it leads to the formation of phases that are instantaneous ground states of the relevant thermodynamic potential, when that thermodynamic potential is extended to include less common variables, such as particle size and environment-dependent surface energy. Beyond a certain stage of growth, the material can be kinetically frozen in an intermediate phase, leading to a bulk-metastable synthesis product[7]. Consequently, metastable phases that are thermodynamic ground states at some intermediate stage of growth are deemed synthetically accessible. This quasi-thermodynamic vision of hydrothermal synthesis provides a baseline of synthesizability that can be readily evaluated computationally to isolate driving forces that favour the formation of a target phase. Knowledge of these driving forces can then guide experimental synthetic efforts, help to computationally select metastable phases that are likely to be synthetically accessible and identify deviations from classical thermodynamic behaviour[8], such as in non-classical nucleation[9].

We base our study of hydrothermal phase selection on the $FeS_2$ mineral system due to its engineering relevance[10,11], geologic importance[12–14] and unresolved structure selection mechanism[13]. The $FeS_2$ system contains two common phases—pyrite and marcasite—and while hydrothermal recipes for the synthesis of both phases are established[12,15–17], the underlying forces governing phase selection during the growth are not understood[13]. It is known that marcasite can be grown as the dominant phase below pH = 5 (refs 12,15,16), despite pyrite being the thermodynamic ground state of bulk $FeS_2$ (ref 18). However,

the mechanism by which pH influences phase selection in $FeS_2$ is unclear as it does not affect the relative stability of bulk pyrite and marcasite[13,15,19,20].

Here we quantify phase selection during the hydrothermal growth of $FeS_2$ by evaluating the full thermodynamic potential governing the evolution of the system throughout the growth process. The thermodynamics describing particle growth in solution are given by the sum of bulk and surface energy, which scales as $\Phi = \frac{4}{3}\pi r^3 g_b + 4\pi r^2 \bar{\gamma}$, where $g_b$ is the volumetric bulk Gibbs free energy of formation, $\bar{\gamma}$ is the particle-averaged surface energy and $r$ is the particle size[8]. Following the formalism outlined above, we propose that the effect of pH can be understood in terms of nucleation and growth from solution that incorporates this competition between bulk and surface stability[21–24]. Contrary to the bulk, the surface energies of the two phases vary with pH due to the adsorption of $H^+$ and $OH^-$ ions[25–27]. By accounting for the adsorption in the evaluation of surface energy, we are able to fully account for the effects of solution chemistry in a theoretical treatment of synthesis, accounting for 'spectator ions' that influence the growth through the surface of the material, but are not represented in the chemical formula of the bulk product. Thus, we evaluate $\Delta\Phi = \Phi_{marcasite} - \Phi_{pyrite}$, the driving force for the formation of the marcasite phase with respect to pyrite, at all stages of growth and as a function of the growth environment. The bulk energy of the growing crystal is determined by the energy of the pure crystal, along with contributions from defect formation, off-stoichiometry and strain. In this work, however, we focus on the growth of pure $FeS_2$ pyrite and marcasite, assuming the bulk energy of both phases to be that of their stoichiometric configuration[28]. The energy of the solid–liquid interface is governed by a combination of bulk-like bond breaking and off-stoichiometry due to adsorption and segregation. It is convenient to approximate the interface energy by the sum of a solid–solvent interface energy and the free energies of adsorption for solute species within the electrostatic double layer, neglecting in this case segregation from the bulk solid. Thus, we separately compute the free energy of a pristine interface between the stoichiometric solid and solvent, $(G^{surface + solvent} - G^{bulk})$, and the free energy of adsorption of solutes from the solution, $\Delta\mu_i^{ads.}$, giving us all the information necessary to obtain the free energy of the solid–liquid interface, $\gamma A = (G^{surface + solvent} - G^{bulk}) + \sum N_i^{ads.} \cdot \Delta\mu_i^{ads.}$. While in principle, adsorption-induced segregation can be included[29], we do not include this coupling for $FeS_2$ as no significant segregation is to be expected in this compound. By evaluating the

---

**Table 1 | Surface and adsorption energetics of $FeS_2$ in water.**

| Phase | Facet | $\gamma_{(hkl)}^{vac}$ | $\gamma_{(hkl)}^{solv}$ | $\Delta E_{H_3O^+}^{ads,\infty} - \Delta E_{slab}^{solv}$ | $\Delta E_{OH^-}^{ads,\infty} - \Delta E_{slab}^{solv}$ |
|---|---|---|---|---|---|
| | | (J m$^{-2}$) | (J m$^{-2}$) | (eV/pH = 0, 473 K) | (eV/pH = 0, 473 K) |
| Pyrite | (100) | 1.38 | 1.11 | 0.300 | 0.789 |
| | (110) | 2.14 | 1.26 | 0.757 | 0.206 |
| | (111) | 1.80 | 1.71 | 0.072 | 0.353 |
| | (210) | 1.82 | 1.07 | 0.899 | 0.299 |
| Marcasite | (001) | 1.70 | 1.45 | 0.215 | 0.210 |
| | (010) | 1.54 | 1.10 | 0.485 | 0.884 |
| | (100) | 2.12 | * | * | * |
| | (011) | 1.75 | 1.74 | 0.097 | −0.246 |
| | (101) | 1.07 | 0.94 | 0.684 | 0.215 |
| | (110) | 1.68 | 1.19 | 0.471 | −0.254 |
| | (111) | 1.67 | 1.21 | 0.443 | −0.097 |

Calculated surface energies of various facets of pyrite and marcasite in vacuum ($\gamma_{(hkl)}^{vac}$) and in contact with pure non-dissociated water ($\gamma_{(hkl)}^{solv}$), as well as the calculated adsorption energy of $H_3O^+$ and $OH^-$ at infinite dilution, with respect to the chemical potential of each ion in solution at pH = 0, 473 K and a hydrated adsorption site. More precisely, $\Delta E^{ads,\infty} - \Delta E_{slab}^{solv}$ captures the strength of the adsorbate–solid interactions with respect to the free energy of the ion in solution and a hydrated solid surface, but does not include the contribution of adsorbate–adsorbate interactions or configurational entropy on the surface. *, Omitted due to convergence issues on the hydration reference state.

thermodynamics relevant to FeS$_2$ particle growth from first principles, we find that the transition from pyrite to marcasite growth under acidic conditions may be explained by the pH-dependent stability of the surfaces of the two phases, suggesting that the quasi-thermodynamic vision of synthesis described here may serve as a valid and computationally accessible metric of the synthesizability of metastable materials.

## Results

**Surface thermodynamics**. We first evaluate the relative energies of the various crystallographic facets in pyrite and marcasite and their tendency to adsorb OH$^-$ and H$_3$O$^+$ ions, as given in Table 1 and illustrated in Fig. 1. In pyrite, the (100) and (210) facets are dominant in the vacuum and solvated cases, in line with the common occurrence of these facets in natural cubic and pyritohedral habits of pyrite[30,31]. In marcasite, we find that the spread of surface energies between the different facets is smaller than in pyrite, with the (010), (101), (110) and (111) facets all having low energies, in agreement with their prevalence in natural marcasite crystals[31]. We then calculate the particle-averaged surface energy of each phase, which gives the total energetic contribution of the surface to the free energy of the solid (Fig. 1b). One way to verify the accuracy of the surface energy curve is through the experimentally measured isoelectric point, which corresponds to a transition from a positively charged (clean or H$_3$O$^+$ adsorbed) to a negatively charged (OH$^-$ adsorbed) surface. The onset of OH$^-$ adsorption onto surface Fe sites in pyrite around pH = 1, seen as the point at which the surface energy of pyrite begins to decrease, agrees well with the experimentally measured isoelectric point (IEP) in pyrite at pH = 1.4 (refs 25,26). Similarly, the absence of a maximum in surface energy in marcasite down to pH = 0 suggests that some of the marcasite surfaces are always hydroxylated, even at very low pH, which agrees with the lack of an experimentally observed IEP in marcasite within an experimentally accessible pH range[32]. In both cases, the agreement of the equilibrium particle morphology and adsorption character of negative ions with experimental observations of the IEP indicates that the interface free energies of the low-energy facets in the OH$^-$-adsorption regime are captured reasonably accurately.

From the particle-averaged surface energies given in Fig. 1b, it is clear that while marcasite surfaces are more stable than that of pyrite under highly acidic conditions, a rapid onset of OH$^-$ adsorption onto pyrite under more basic conditions stabilizes the pyrite surface. The origin of this transition lies in the variation in OH$^-$ adsorption strength among the various facets of pyrite and marcasite. In pyrite, OH$^-$ adsorbs onto the (210) facets covering the majority of the Wulff shape, while in marcasite, stabilization due to OH$^-$ adsorption is initially limited to the otherwise unstable (110) and (111) facets.

**Phase selection during synthesis**. The influence of surface energy on phase selection in FeS$_2$ synthesis can be viewed from both a thermodynamic and a kinetic standpoint. Combining the bulk and surface energy of pyrite and marcasite across all sizes and pH levels, we construct the size–pH phase diagram of FeS$_2$, given in Fig. 2. We can immediately see that marcasite is the lowest-energy phase in acid at small particle sizes, giving rise to a driving force for the formation of marcasite under these conditions. At this stage of growth, the system is significantly influenced by nucleation kinetics, which we can analyse within the scope of classical nucleation theory. As shown schematically in Fig. 3, nucleation from solution proceeds over a nucleation barrier, which arises from the energetic penalty of forming a high-surface-area critical nucleus and scales as $\bar{\gamma}^3/g_b^2$, where $\bar{\gamma}$ is the average surface energy of the nucleating phase and $g_b$ is the volumetric bulk-driving force for precipitation[8]. The relative rates of pyrite and marcasite nucleation are exponential in the difference between their nucleation barriers, such that even a small decrease in surface energy from pyrite to marcasite can lead to a large excess of marcasite nucleation. Taking the experimentally reported supersaturation for FeS$_2$ hydrothermal growth[15], we can immediately see that at the critical nucleus size, marcasite has a lower free energy than pyrite below pH = 4, and thus nucleates exponentially faster. This transition to marcasite-dominant nucleation agrees with the experimentally observed onset of marcasite formation between pH = 4 and pH = 6 (refs 12,15), and the absence of marcasite when FeS$_2$ is grown under more basic conditions. Thus, we can conclude that marcasite growth in acidic media may be explained by its finite-size thermodynamic stability and preferential nucleation under these conditions.

## Discussion

The agreement between the experimentally observed stabilization of marcasite in acid and our computational results lends significant credibility to the model of synthesis derived here. Considering that the energy scale of pH is small compared with typical energy scales involved in solid-state chemistry, the

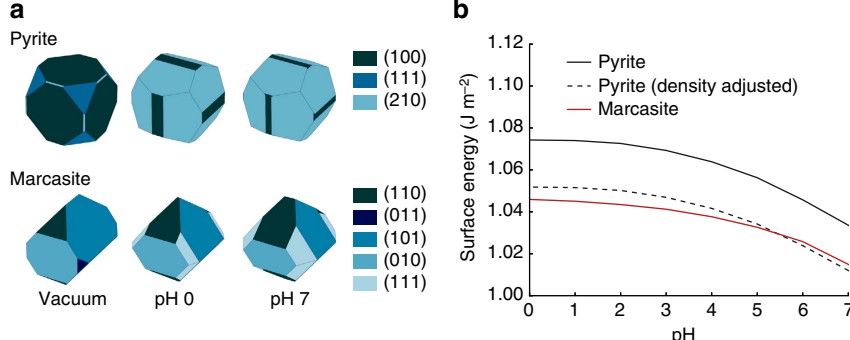

**Figure 1 | Surface energies of FeS$_2$ pyrite and marcasite.** (**a**) Equilibrium particle shapes (Wulff shapes) for pyrite and marcasite in vacuum and in solution at pH = 0 and pH = 7. (**b**) Surface energies of pyrite and marcasite averaged over the equilibrium Wulff shape across a range of pH levels. The solid lines give the surface energy of pyrite and marcasite per unit area, while the dashed line gives $\bar{\gamma}_p(\frac{\rho_m}{\rho_p})^{2/3}$, the effective molar surface energy of pyrite scaled to account for the higher density of pyrite relative to that of marcasite. The density adjustment is necessary for a direct comparison of the effect of surface energy on stability, as it accounts for the fact that the relevant free energy for determining the relative stability of pyrite and marcasite is the molar free energy. Thus, the surface energy of a pyrite particle with an equal mole number to that of a marcasite particle must be scaled down to account for the smaller size of the denser pyrite structure.

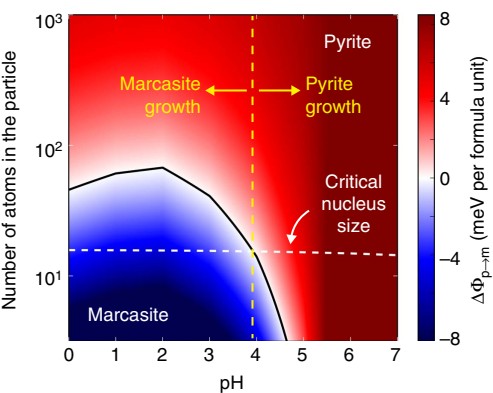

**Figure 2 | Thermodynamics of nanoscale FeS₂.** The finite-size phase diagram of FeS₂ across a range of pH values, illustrating the low-particle-size, low-pH region of thermodynamic stability for marcasite. Note that we report a single critical nucleus size based on the experimentally reported supersaturation[15] for both pyrite and marcasite because the difference between the two is negligible.

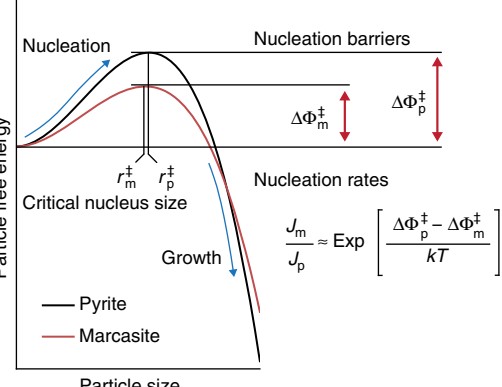

**Figure 3 | Relationship between finite-size energetics and nucleation kinetics.** A schematic illustration of nucleation kinetics leading to the formation of metastable marcasite due to a lower kinetic barrier to nucleation $\Delta\Phi_m^{\ddagger}$ and a correspondingly exponentially higher nucleation rate $J_m$, relative to that of pyrite ($\Delta\Phi_p^{\ddagger}$ and $J_p$, respectively)[8].

agreement of the transition point between pyrite and marcasite within 1–2 pH units is quite remarkable. We speculate that the primary reason underlying this result is the systematic cancellation of error between the chosen adsorbed and reference states. For example, we neglect dispersion interactions in our model due to computational constraints, despite the fact that they certainly play a significant role in determining the behaviour of the real FeS₂-water interface. However, the error due to this necessary simplification likely cancels between the adsorbed and reference states of the ions, giving a sufficiently accurate estimate of the behaviour of the system.

One other potential issue in the analysis presented here is the validity of the classical energy decomposition into bulk and surface terms to obtain the energy of the nucleus[8]. Accounting for the exact dynamics and free energies of the growing nucleus is extraordinarily difficult given any current computational or experimental method, and impractical given the goal of obtaining a scalable 'synthesizability filter' for computational materials discovery. Instead, the semi-continuum analysis presented here aims to provide a first-order extrapolation of finite-size free energies from the bulk to the size scales relevant to nucleation. Of course, given the small energy scales involved, more detailed studies of the small-scale thermodynamics and nucleation kinetics in this system would help clarify the validity of the approximations made here, as well as identify any non-classical nucleation and growth behaviour that may occur.

We expect that the approach used here to study phase selection between pyrite and marcasite FeS₂ can be widely applied to studying polymorphism in other chemical systems synthesized by the hydrothermal method. Indeed, finite-size stability of CaCO₃ mediated by Mg uptake from solution has been used to explain the preferential nucleation of metastable aragonite over calcite in present-day oceans[24]. The general lack of empirical parameters in the derivation of the finite-size phase diagram allows this approach to be extended to chemical spaces with scarce to no experimental data, as is often necessary for computational materials discovery. By constructing the finite-size phase diagram with the inclusion of any arbitrary set of 'spectator ions' adsorbing on the solid surface, it is possible to identify approximate solution conditions under which there may be a driving force for the formation of a target metastable phase. With this knowledge, one can design syntheses that would allow the system to express the identified driving force, nucleating within the desired region of the phase diagram. Thus, we believe that

both the general model proposed here, and the analysis of FeS₂ can serve as a useful thermodynamic baseline for predicting phase selection during synthesis and assist the realization of novel materials.

## Methods

**Thermodynamic model of an aqueous interface.** The defining feature of an aqueous interface is the existence of an electrostatic double layer due to the adsorption of charged species to the solid[33]. However, much of the complexity of the double layer can be neglected when calculating the total energy of the interface. To derive an approximate treatment of this structure, it is helpful to break down the electrostatic double layer into three components: the electronic 'space charge' region, the chemisorbed region within the Helmholtz plane and the physisorbed 'diffuse' region outside the Helmholtz plane, as shown schematically in Fig. 4a. The adsorption energy of chemisorbed species is likely large, and thus must be represented accurately, accounting among other features for the charge state of the adsorbate. The space charge region in the solid arises due to the changes in the electronic structure of the solid associated with bond breaking and adsorption from the liquid, and thus will be captured intrinsically by any accurate first-principles model of the surface and the chemisorbed layer. In contrast, the contribution of the diffuse layer to the total energy of the system is not always significant. The electrostatic potential at the Helmholtz plane, experimentally measured as the $\xi$ potential, is typically on the order of 40 mV (refs 25,26), while the capacitance of the double layer region is on the order of 0.5 F m⁻² (refs 25,26), meaning that the total energy stored in the diffuse layer is on the order of 10⁻⁴ J m⁻². This quantity is negligible on the scale of total interfacial energies in ceramic–aqueous systems, which are in the range of 1 J m⁻² (ref. 34). Thus, a model of the chemisorbed species within the Helmholtz plane, constructed to ensure the correct charge state of the adsorbed species, but neglecting the details of the diffuse layer, yields an accurate estimate of the true interfacial energy, even in the presence of an electrostatic double layer.

To fully model a solid–liquid interface, it is generally necessary to consider the adsorption of all ions present in solution. A more tractable simplification is to consider only the effect of known potential-determining ions, as these ions are by definition those which adsorb strongest and thus determine the structure of the double layer. In the case of FeS₂, the potential-determining ions are $H_3O^+$ and $OH^-$ (refs 25,26), suggesting that in an aqueous medium, the tightly bound chemisorbed layer consists primarily of these species, in addition to the $H_2O$ solvent molecules. Thus, it is only necessary to consider the adsorption of $OH^-$, $H_2O$ and $H_3O^+$, accounting for all other ions only to the extent that they set the ionic strength and pH of the solution. The energy of adsorbing $H_2O$ is equivalent to the energy of solvating the solid surface, and will be addressed in a later section. Further, we assume that at a given pH, $OH^-$ and $H_3O^+$ will never be adsorbed simultaneously. On the basis of this set of assumptions, the adsorption energy of the two ions can be written based on the minimization of the free energy of adsorption, given by the enthalpy $\Delta H^{ads}$ and entropy $\Delta S^{ads}$ of adsorption with respect to the number of adsorbates $N^{ads}$:

$$N_{H_3O^+}^{ads}\,\Delta\mu_{H_3O^+}^{ads} = \min_{N_{H_3O^+}^{ads}}\left[\Delta H_{H_3O^+}^{ads} - T\Delta S_{H_3O^+}^{ads}\right]$$

$$N_{OH^-}^{ads}\,\Delta\mu_{OH^-}^{ads} = \min_{N_{OH^-}^{ads}}\left[\Delta H_{OH^-}^{ads} - T\Delta S_{OH^-}^{ads}\right]$$

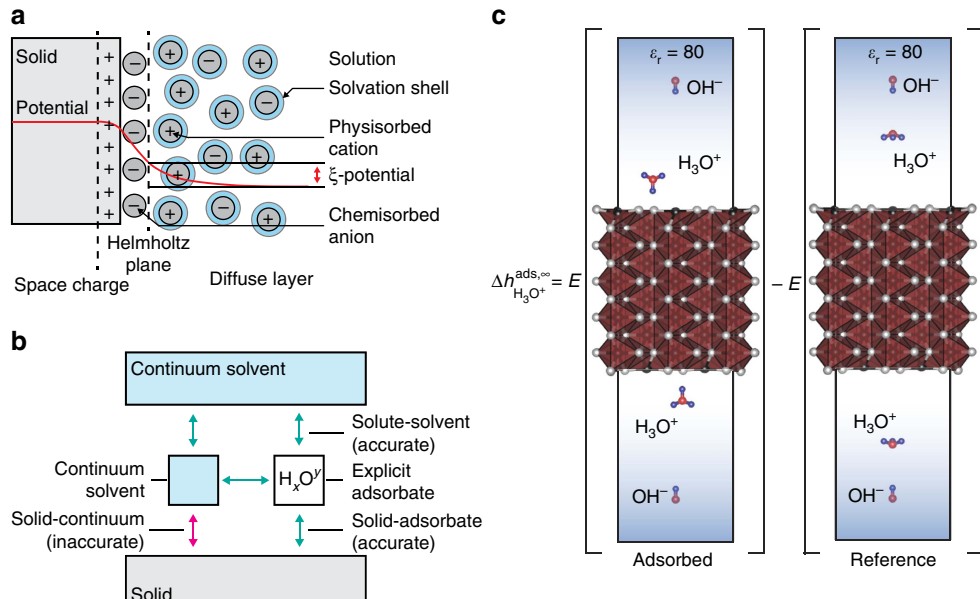

**Figure 4 | Scheme for computing the free energy of an aqueous interface. (a)** The structure of an electrostatic double layer, consisting of a space charge in the solid, a tightly bound chemisorbed layer within the Helmholtz plane and weakly bound physisorbed ions in the diffuse layer. (**b**) The continuum solvation model provided by VASPsol, which accurately captures the interactions between solute molecules and the solvent, as well as the solvent with itself, but does not accurately represent the interactions between the solvent and an extended solid surface. We introduce a solvation scheme to correct the unreliable solid–solvent interaction with a more physical model calculated from the explicit adsorption of water molecules. (**c**) To calculate the adsorption energy of charged ions at infinite dilution, we choose a calculation scheme that allows electron transfer between the cationic and anionic species to occur self-consistently. In both the adsorbed and reference states, the ions are sufficiently separated to allow the continuum solvent to partially screen the interactions between them, such that the effect of the electrostatics can be subtracted out analytically as a post-calculation correction. Here we show the adsorption of $H_3O^+$ onto a sulfur site on the marcasite (001) surface as an example of this calculation, within the VASPsol continuum solvent model.

We approximate the enthalpy of adsorption $\Delta H^{ads}$ by accounting for adsorbate–solid, adsorbate–solvent and adsorbate–adsorbate interactions. The adsorbate–solid and adsorbate–solvent interactions are captured by the enthalpy of adsorption at infinite dilution $\Delta h^{ads,\infty} = E^{ads} - E^{ref}$, where $E^{ads}$ and $E^{ref}$ are the density functional theory (DFT) energies of the ion adsorbed onto the solid and in solution respectively, shown schematically in Fig. 4c. Note that this energy of adsorption includes the energy of desorbing a water molecule, as the adsorption process is competitive with the pure solvent. We approximate adsorbate–adsorbate interactions with the Debye–Huckel model of screened electrostatics in an electrolytic medium, given here by $V^{el}$. Thus, we can write the enthalpy of adsorption for $OH^-$ and $H_3O^+$ as:

$$\Delta H^{ads}_{H_3O^+} = N_{H_3O^+}\left(E^{ads}_{H_3O^+} - E^{ref}_{H_3O^+}\right) + V^{el}$$

$$\Delta H^{ads}_{OH^-} = N_{OH^-}\left(E^{ads}_{OH^-} - E^{ref}_{OH^-}\right) + V^{el}$$

$$V^{el} = \frac{1}{2}\sum_{i\neq j}\frac{q_i q_j e^{-|r_i-r_j|/\lambda}}{4\pi\varepsilon_r\varepsilon_0 \ |\ r_i-r_j\ |}$$

where $\mathbf{r}_i$ are the positions of the adsorbed ions on the solid surface, $\lambda$ is the Debye screening length of the solution and $\varepsilon_r$ is the dielectric constant of the solution near the interface. In our model, we use a Debye screening length of $\lambda = 1.0$ nm, based on an average over screening lengths for reported synthesis recipes for pyrite and marcasite[15]. As we are considering the two-dimensional electrostatic interactions between adsorbates, screened by adsorbed water molecules, we set the dielectric constant $\varepsilon_r = 12$, estimated from reported experimental and computed values of the dielectric constant of interfacial water in similar systems[35,36]. Finally, we set our temperature to 473 K in accordance with the experimental conditions commonly reported for $FeS_2$ hydrothermal growth[15,16].

To obtain the entropy of adsorption, we consider the entropy of the adsorbed and solution states of the ion, $s^{ads}$ and $s^{soln}$, respectively. The entropy of the adsorbed ion is well approximated by the configurational entropy over adsorption sites. The entropy of the ion in solution is given by the configurational entropy over the translational degrees of freedom of the ion in solution, which, assuming that $OH^-$, $H_2O$ and $H_3O^+$ all have approximately the same volume, is given by $k_B\log[x]$, where $x$ is the mole fraction of the ion of interest. We then relate the entropy of $H_3O^+$ to pH by treating pH as an activity with respect to a 1 M solution of $H_3O^+$ at standard state and assuming that the solution behaves ideally, which yields:

$$s^{soln}_{H_3O^+} = k_B\frac{T_0}{T}\ln M_w + 2.3k_B\,pH$$

where the $M_w$ is the molarity of water and $T_0 = 300$ K is temperature in the reference state. Following the same assumptions, as well as the fact that $OH^-$, $H_2O$ and $H_3O^+$ are in equilibrium, we derive the entropy of $OH^-$ in solution in terms of the calculated formation enthalpy of $H_3O^+$ and $OH^-$ from $2H_2O$, which we denote $\Delta h^0_w$:

$$s^{soln}_{OH^-} = \frac{\Delta h^0_w}{T} - k_B\frac{T_0}{T}\ln M_w - 2.3k_B\,pH$$

A detailed derivation of these results is given in Supplementary Note 1. Combining the solution references with the configurational entropy of the adsorbed state, we have the entropy of adsorption for both $H_3O^+$ and $OH^-$ in terms of pH:

$$\Delta S^{ads}_{H_3O^+} = N_{H_3O^+}\left(s^{ads}_{H_3O^+} - s^{soln}_{H_3O^+}\right)$$

$$\approx N_{H_3O^+}\left(s^{ads}_{H_3O^+} - 2.3k_B\,pH - k_B\frac{T_0}{T}\ln M_w\right)$$

$$\Delta S^{ads}_{OH^-} = N_{OH^-}\left(s^{ads}_{OH^-} - s^{soln}_{OH^-}\right)$$

$$\approx N_{OH^-}\left(s^{ads}_{OH^-} - \frac{\Delta h^0_w}{T} + 2.3k_B\,pH + k_B\frac{T_0}{T}\ln M_w\right)$$

We have thus obtained a thermodynamic picture of ion adsorption that is efficiently computable from first principles and captures the primary trends we could expect to see at the solid–liquid interface as a function of pH. A similar analysis can be readily performed for other dissolved ions, based on their computed solubility product $K_{sp}$, generalizing this approach to a solid–aqueous interface with any ideal or near-ideal aqueous solution.

**Computational implementation.** On the basis of the thermodynamic framework derived above, it is clear that to obtain a full quasi-thermodynamic picture of the hydrothermal growth of $FeS_2$ pyrite and marcasite, only a few density functional theory calculations are necessary. First, we must calculate the bulk energy and structure of pyrite and marcasite. Then, for each low-energy crystallographic facet of each phase we must obtain the interfacial energy between the $FeS_2$ solid and water, or equivalently, the solvation energy of each crystal facet, $(G^{surface + solvent} - G^{bulk})$. Finally, for each solvated facet, we must calculate the enthalpy of adsorbing dilute $H_3O^+$ and $OH^-$ ions onto all likely adsorption sites, $\Delta h^{ads,\infty}$. An example calculation, illustrating the thermodynamic formalism can be found in Supplementary Note 2.

**Computational details.** All calculations were done using the Vienna *Ab-Initio* Simulation Package (VASP)[37,38] implementation of DFT, using projector augmented wave (PAW) pseudopotentials[39,40] with a plane-wave basis set using an energy cutoff of 520 eV. Consistently with previously reported results, we find that the PBEsol exchange-correlation functional[41] provides an accurate and computationally efficient model of $FeS_2$ (refs 42,43), correctly stabilizing pyrite over marcasite as the ground state of the system in agreement with experiment[12,44] and higher-order functionals, although not quite reaching the experimentally measured transition enthalpy between the two phases[18] (Supplementary Table 1). To ensure consistency between the high-symmetry bulk calculations and low-symmetry adsorption calculations, we remove all symmetry restrictions from the calculation, giving the system identical relaxation degrees of freedom across all calculations. Finally, for bulk calculations, we choose a Γ-centred k-point mesh ($6 \times 6 \times 6$ for pyrite and $6 \times 6 \times 8$ for marcasite) based on previously optimized calculation parameters in similar systems[45].

In our surface calculations, we consider the symmetrically distinct low-index facets of pyrite and marcasite previously reported to be significant. Specifically, in pyrite, we consider the (100), (110), (111) and (210) facets[46,47], while in marcasite, we consider the (100), (010), (001), (110), (101), (011) and (111) facets[31], defined with respect to the unit cells given in Supplementary Table 1. To generate the surface structures, we choose surface terminations that minimize the number and strength of bonds broken, evaluated based on the integral of charge density associated with each bond in question, and are maximally non-polar, following the Tasker surface stability criterion[48]. In the case where several surface terminations satisfy these criteria, we consider all such terminations. The resulting most stable (under solvated conditions) surface structures are shown in Supplementary Fig. 1. While there is limited experimental data available to verify the accuracy of this approach, in the case of the well-characterized pyrite (100) surface, our approach leads to a surface structure consistent with that derived from low-energy electron diffraction (LEED) characterization[49]. Finally, we neglect the contribution of the solid to the solid–liquid interface entropy as it is known to be negligibly small in similar ceramic systems[50].

**Solvation model.** To account for solvation, we rely on the VASPsol continuum solvation model[51] to avoid the computationally prohibitive sampling of explicit solvent configurations. The VASPsol model serves two important purposes–it reproduces the mean-field interactions between ions and bulk solvent, and provides a dielectric medium that screens electrostatic interactions between charged adsorbates, their counterions and their periodic images. However, while the VASPsol model is known to accurately reproduce the energy of solvating isolated molecules[51], its performance with respect to the solvation of solid surfaces is uncertain.

To correct any unphysical interactions between the VASPsol continuum solvent and the solid slab, we introduce a solvation correction scheme. We assume that the VASPsol model accurately reproduces all solvent–solvent and solvent–ion interactions, but fails to capture solvent–solid interactions, as shown schematically in Fig. 4b. To correct this error, we first remove the energy associated with the interaction of the continuum solvent and the solid by subtracting out the difference between the energy of the clean surface (solid 'slab') in contact with vacuum $E_{slab}^{vac}$ and in contact with the continuum solvent $E_{slab}^{vaspsol}$, which we will refer to as $\Delta E_{slab}^0 = E_{slab}^{vaspsol} - E_{slab}^{vac}$. We then add back the interactions between the solvent and the slab by explicitly calculating the energy of adsorbing isolated water molecules within the continuum solvent, $\Delta E_{slab}^{solv} = E_{H_2O,ads}^{vaspsol} - (E_{H_2O,ref}^{vaspsol} - TS_{H_2O,ref}^{exp.})$, for each adsorption site on the solid, where $S_{H_2O,ref}^{exp.}$ is the experimentally measured entropy of bulk water. Note that we neglect the entropy of the adsorbed water as we assume that the interfacial water layer is relatively constrained and ice-like, significantly reducing its entropy relative to that of the bulk solution[52]. Having obtained this shift for each adsorption site, we can correct any calculation done with only the continuum solvent to capture the solid–solvent interactions potentially misrepresented by VASPsol.

For example, to calculate the energy $E_{interfac}$ of a surface with $N^{sites}$ identical adsorption sites, of which $N^{ads}$ are occupied by some adsorbing ions and $N^{sites} - N^{ads}$ are filled by water, we calculate the energy of a periodic slab with explicit adsorbed ions (but not water molecules) within VASPsol to get $E_{interface}^{vaspsol}$. We then apply the solvation correction to get the true interface energy:

$$E_{interface} = E_{interface}^{vaspsol} - \Delta E_{slab}^0 + (N^{sites} - N^{ads})\Delta E_{slab}^{solv}$$

In the case where there are distinct adsorption sites, the energy of solvation $E_{slab}^{solv}$ becomes site specific. For $FeS_2$, we assume that the site-specific solvation energy is determined by the local chemistry, giving separate solvation energies for Fe and S sites on the surface. As it is impossible to adsorb $H_2O$ simultaneously to adjacent Fe and S sites due to steric constraints, we take the lower energy of the Fe and S adsorption sites for each facet as the facet-specific solvation energy $\Delta E_{slab}^{solv}$, and the density of these sites as the number of adsorption sites $N^{sites}$. Combining these terms, we obtain the solid–solvent interfacial energy term, with $N^{ads}=0$:

$$(G^{surface + solvent} - G^{bulk}) \approx E_{interface}^{vaspsol} - \Delta E_{slab}^0 + N^{sites}\Delta E_{slab}^{solv} = E_{slab}^{vac} + N^{sites}\Delta E_{slab}^{solv}$$

where $E_{interface}^{vaspsol} - \Delta E_{slab}^0$ simplifies to $E_{slab}^{vac}$ in the case of zero adsorbed ions.

**Charged adsorption.** To obtain a full picture of interfacial stability across various solution conditions (here, pH levels), it is necessary to calculate the enthalpy of adsorption of all relevant ions (here, $OH^-$ and $H_3O^+$) at infinite dilution, as discussed in the thermodynamic derivation earlier. To do so, it is necessary to ensure that in both the adsorbed and reference state, the charge state of the adsorbing ion is physical. Under periodic boundary conditions imposed by plane-wave DFT, the charge state can be set explicitly by removing a number of electrons from the system, and compensating the resulting charge with a homogeneous background. A more physical model is to include a counter charge in the system and allow charge transfer from the cationic to the anionic species to occur self-consistently. However, in this case, it is important to take care to ensure that no unphysical electrostatic interactions between the anion, cation and their periodic images contribute to the adsorption energy.

One approach to ensure that unphysical electrostatic interactions do not contribute to the adsorption energy is to construct a supercell large enough, such that the electrostatic interactions between ions decay to a negligible level. A more computationally tractable approach is to choose a reference state, such that the electrostatic interactions either cancel out between the adsorbed and reference states, or can be subtracted out analytically, giving an accurate enthalpy of adsorption at infinite dilution $\Delta h_{H_3O^+}^{ads,\infty} = E_{H_3O^+}^{ads} - E_{H_3O^+}^{ref}$. One such choice of reference state is given in Fig. 4c. In this set-up, the nearest neighbour cation-cation and anion-anion image interactions cancel out between the adsorbed and reference state, leaving only the cation-anion interactions and ion–solid interactions. In both cases, the continuum solvent medium provided by VASPsol ensures rapid decay of the electrostatic interactions as a function of distance, such that even at a 5 Å minimum separation (with a VASPsol dielectric constant $\varepsilon_r = 80$ for an aqueous system), electrostatic interactions between the ions are small enough that they can be subtracted out from the total energy as a post-calculation correction. If we assume that the ion–solid interaction in the reference state is small, which is reasonable considering that the solid is not charged and the separation between the ion and solid is over 10 Å, the only remaining interaction is the one we are interested in—the ion–solid interaction in the adsorbed state. Note that in this case, the ion charge state is self-consistently set to the correct value in both the adsorbed and reference configurations, as can be confirmed by a Bader charge analysis[53]. Relaxed adsorption geometries for $H_3O^+$ and $OH^-$ derived using this approach for all facets of pyrite and marcasite can be found in Supplementary Figs 2 and 3, respectively.

**Data availability.** Complete computational data to support the findings of this study is available from the authors on reasonable request.

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

## Acknowledgements

We thank Wenhao Sun, Pieremanuele Canepa and Sai Jayaraman for fruitful discussions. Support for this work was provided by the NSF Software Infrastructure for Sustained Innovation (SI2-SSI) Collaborative Research program of the National Science Foundation under Award No. OCI-1147503. Computational resources for this project were provided by the National Energy Research Scientific Computing Center, a DOE Office of Science User Facility supported by the Office of Science of the US. Department of Energy under contract no. DE-AC02-05CH11231.

## Author contributions

D.A.K. and G.C. designed the study. D.A.K. performed computational experiments, analysed the data and prepared the manuscript. All authors discussed the results and commented on the manuscript.

## Additional information

**Competing financial interests:** The authors declare no competing financial interests.

