## [Peer Review File · Nature Communications]

Reviewers' comments:

Reviewer #1 (Remarks to the Author):

Structure selection during hydrothermal growth from first-principles: the case of FeS₂ pyrite and marcasite by Kitchaev and Ceder.

I found this paper well-written and clear. It provides a valuable framework for predicting stability of metastable materials making the paper interesting to a broad community. The methodology and calculations seem to be well reported, apart from giving an example, which I mention below.

The calculation of the variation of surface energy with pH and the effect of both pH and crystallite size on phase stability and of interest to a wide range of communities from pharmaceuticals to geochemistry. On the FeS₂ system, this paper has made an excellent contribution of providing an explanation of why pyrite is preferred at high pH.

I consider that the paper is worthy of publication in Nature Comm. I have only a few minor points that need clarification for the interested reader to follow completely.

i. In Figure 1, the authors refer to the scaled surface energy of pyrite for higher density. I was not sure how this was achieved, and hence I do not follow the argument that OH⁻ stabilises pyrite. As well as not understanding how and the normalization of the pyrite surface is done it is not clear to me how it is connected later with the calculations of the adsorption energies and stabilities.

ii. Why does a lack of a maximum in surface energy agree with the lack of an experimentally observed isoelectric point in the specified pH range?

iii. The SI gives a nice description of the calculation approach with all of the various terms. However, it does not give any insight as to how useful or important these terms are and therefore it is not possible to gauge the merit of including all of these terms without repeating the calculation. I ask that at least one example be given, say at pH 5.5 of the energies and entropies, so that the reader can also judge the value of each part of the calculation. This would be more useful than the lattice parameters, which are all well documented.

iv. For a similar reason please would the authors add in SI optimized adsorption geometries for the most stable facets with adsorbates as there is only schematic representation of these.

Reviewer #2 (Remarks to the Author):

Among many semiconducting materials like, CuO, Cu₂O, Cu₂S, etc., FeS₂ (iron pyrite) has received significant attention because of its low cost and high availability. In addition, good semiconducting properties and optical absorption properties of Iron Pyrite makes it an attractive material for the solar energy conversion. However, the poor open circuit voltage of Iron Pyrite largely reduces its efficiency in converting the solar energy into electrical energy. The surface defects, impurity and stoichiometry are the main reasons behind the low performance of iron pyrite materials.

In this study, the authors report the PH effect on the phase selection of FeS₂ (Marcasite or Pyrite) during the hydrothermal synthesis. The PH effect on the FeS₂ formation (experimentally) has been reported in the literatures, and the underline mechanism is unknown in the past. Therefore, this study indeed is beneficial for this aspect in FeS₂ formation using the hydrothermal method. As the iron pyrite is sensitive to the moisture (degradation) or environmental conditions, the hydrothermal approach for the iron pyrite could be detrimental for high-quality pyrite phase formation. The justification for the modeling of hydrothermal iron pyrite is needed for this study, which could also broaden the impact on the growth of iron pyrite. Furthermore, the PH effect on the phase selection could be a good subject for the modeling efforts, therefore, a universal material predictive model will have broader impact (iron pyrite itself (or CaCO₃) may not be enough to guarantee this).

As the conclusion, the paper contains a significant amount of work done in modeling the PH effect on hydrothermally grown iron pyrite materials. It also contains the work to prove the PH effect in the previous experimental efforts. Having no universal predictive model on phase selection makes it an incomplete paper.

Reviewer #3 (Remarks to the Author):

The manuscript reports a model, based on ab initio thermodynamic methods, for the phase selection of pyrite vs marcasite phases of FeS₂ produced by hydrothermal methods. It is shown that the experimentally observed relationship between crystal phase and pH can be explained by a model which takes the surface energies of small FeS₂ crystals with adsorbed OH⁻ and H₃O⁺ species into account. The model correctly predicts the transition to within 2 pH units. Application of the model to other systems is discussed.

The problem considered is of genuine interest, and the computational methods chosen appear to

be appropriate. The approximations made in the model appear to be reasonable. The paper is well written and clear for the reader (although it is a shame that so many of the details of the method are confined to the supplementary material). I believe that the originality of the model and the potential for the applicability of the methods developed here to a broader range of problems are sufficient to make it suitable for publication in Nature Communications.

Response to reviewers:

#####

Reviewer #1

Structure selection during hydrothermal growth from first-principles: the case of FeS₂ pyrite and marcasite by Kitchaev and Ceder.

I found this paper well-written and clear. It provides a valuable framework for predicting stability of metastable materials making the paper interesting to a broad community. The methodology and calculations seem to be well reported, apart from giving an example, which I mention below.

The calculation of the variation of surface energy with pH and the effect of both pH and crystallite size on phase stability and of interest to a wide range of communities from pharmaceuticals to geochemistry. On the FeS₂ system, this paper has made an excellent contribution of providing an explanation of why pyrite is preferred at high pH.

I consider that the paper is worthy of publication in Nature Comm. I have only a few minor points that need clarification for the interested reader to follow completely.

i. In Figure 1, the authors refer to the scaled surface energy of pyrite for higher density. I was not sure how this was achieved, and hence I do not follow the argument that OH⁻ stabilises pyrite. As well as not understanding how and the normalization of the pyrite surface is done it is not clear to me how it is connected later with the calculations of the adsorption energies and stabilities.

ii. Why does a lack of a maximum in surface energy agree with the lack of an experimentally observed isoelectric point in the specified pH range?

iii. The SI gives a nice description of the calculation approach with all of the various terms. However, it does not give any insight as to how useful or important these terms are and therefore it is not possible to gauge the merit of including all of these terms without repeating the calculation. I ask that at least one example be given, say at pH 5.5 of the energies and entropies, so that the reader can also judge the value of each part of the calculation. This would be more useful than the lattice parameters, which are all well documented.

iv. For a similar reason please would the authors add in SI optimized adsorption geometries for the most stable facets with adsorbates as there is only schematic representation of these.

#####

We thank the reviewer for evaluating our manuscript and the detailed commentary provided. We address all points of concern below and indicate the relevant changes to the manuscript.

i. *“In Figure 1, the authors refer to the scaled surface energy of pyrite for higher density. I was not sure how this was achieved, and hence I do not follow the argument that OH- stabilises pyrite. As well as not understanding how and the normalization of the pyrite surface is done it is not clear to me how it is connected later with the calculations of the adsorption energies and stabilities.”*

The intent of the adjustment in Fig. 1 is to compare the surface energies of pyrite and marcasite at constant mole number, which is the relevant constraint for determining phase stability at a given point in particle growth. Our calculated surface energies are initially normalized to particle surface area, as surface energies are normally reported per unit area. But when comparing the energy of different phases one needs to work on a per mole basis. Directly comparing the surface energy of pyrite and marcasite neglects the fact that a particle of the denser pyrite phase with some number of moles of FeS₂ will be smaller than a particle of marcasite with the same mole number. To account for the smaller size, and correspondingly smaller surface area, of the pyrite particle, we give a directly comparable effective “molar surface energy” for pyrite, which is mathematically equal to $\bar{\gamma}_p * \left(\frac{\rho_m}{\rho_p}\right)^{2/3}$. This energy term is best interpreted as the surface energy contribution to the total energy of a marcasite to pyrite phase transformation at a fixed mole number, normalized to the surface area of the initial marcasite phase, analogous to the energy expression used, for example, by H. Zheng and J. Banfield in their work on nanocrystalline TiO₂ [1].

To clarify the meaning of the density adjustment in Fig. 1, we have added the following to the caption of Fig. 1:

*“... The solid lines give the surface energy pyrite and marcasite per unit area, while the dashed line gives $\bar{\gamma}_p * \left(\frac{\rho_m}{\rho_p}\right)^{2/3}$, the effective molar surface energy of pyrite scaled to account for the higher density of pyrite relative to that of marcasite. The density adjustment is necessary for a direct comparison of the effect of surface energy on stability as it accounts for the fact that the relevant free energy for determining the relative stability of pyrite and marcasite is the molar free energy. Thus, the surface energy of a pyrite particle with an equal mole number to that of a marcasite particle must be scaled down to account for the smaller size of the denser pyrite structure.”*

[1] Zheng, Hengzhong; Banfield, Jillian. "Thermodynamic analysis of phase stability of nanocrystalline titania." *Journal of Materials Chemistry* 8.9 (1998): 2073-2076.

ii. *Why does a lack of a maximum in surface energy agree with the lack of an experimentally observed isoelectric point in the specified pH range?*

The variation in surface energy with pH is determined by the amount of H₃O⁺ and OH⁻ absorbed, with an increase in pH increasing the energy of an H₃O⁺-covered surface, and decreasing the energy of an OH⁻-covered surface. The absence of a maximum in surface

energy down to a very low pH suggests that even under acidic conditions, some of the surface remains primarily hydroxylated. Similarly, an experimentally-measured isoelectric point corresponds to a transition from the preferential absorption of H_3O^+ to that of OH^- . The lack of an isoelectric point within the experimental pH range suggests that the surface is always hydroxylated, in agreement with the lack of a maximum in surface energy.

We have edited the following section to the manuscript in clarify this point:

“One way to verify the accuracy of the surface energy curve is through the experimentally-measured isoelectric point, which corresponds to a transition from a positively-charged (clean or H_3O^+ -adsorbed) to a negatively-charged (OH^- -adsorbed) surface. The onset of OH^- adsorption onto surface Fe sites in pyrite around $pH=1$, seen as the point at which the surface energy of pyrite begins to decrease, agrees well with the experimentally measured isoelectric point (IEP) in pyrite at $pH=1.4$. Similarly, the absence of a maximum in surface energy in marcasite down to $pH=0$ suggests that some of the marcasite surfaces are always hydroxylated even at very low pH, which agrees with the lack of an experimentally observed isoelectric point in marcasite within an experimentally accessible pH range.”

iii. The SI gives a nice description of the calculation approach with all of the various terms. However, it does not give any insight as to how useful or important these terms are and therefore it is not possible to gauge the merit of including all of these terms without repeating the calculation. I ask that at least one example be given, say at $pH 5.5$ of the energies and entropies, so that the reader can also judge the value of each part of the calculation. This would be more useful than the lattice parameters, which are all well documented.

We have added a pair of example calculations to the Supplementary Data, deriving, as an example, the adsorbed surface energies of pyrite (210) and marcasite (110) at $pH 5.5$. The derivation illustrates that all the terms included in the thermodynamic formalism (vacuum surface energy, hydration energy, dilute adsorption energy, Debye-Huckel repulsion, configurational entropy) are significant and hopefully clarifies the calculation procedure for the reader. We have also added the DFT-calculated values necessary to reproduce this calculation for all other facets to the Supplementary Data, Table S2.

iv. For a similar reason please would the authors add in SI optimized adsorption geometries for the most stable facets with adsorbates as there is only schematic representation of these.

We have added a figure illustrating all low-energy adsorption geometries to the Supplementary Data, Figure S3. We hope this figure clarifies which adsorption sites we find to be lowest energy for both H_3O^+ and OH^- , and the configuration of the adsorbate within that site.

#####

Reviewer #2

Among many semiconducting materials like, CuO, Cu₂O, Cu₂S, etc., FeS₂ (iron pyrite) has received significant attention because of its low cost and high availability. In addition, good semiconducting properties and optical absorption properties of Iron Pyrite makes it an attractive material for the solar energy conversion. However, the poor open circuit voltage of Iron Pyrite largely reduces its efficiency in converting the solar energy into electrical energy. The surface defects, impurity and stoichiometry are the main reasons behind the low performance of iron pyrite materials.

In this study, the authors report the PH effect on the phase selection of FeS₂ (Marcasite or Pyrite) during the hydrothermal synthesis. The PH effect on the FeS₂ formation (experimentally) has been reported in the literatures, and the underline mechanism is unknown in the past. Therefore, this study indeed is beneficial for this aspect in FeS₂ formation using the hydrothermal method. As the iron pyrite is sensitive to the moisture (degradation) or environmental conditions, the hydrothermal approach for the iron pyrite could be detrimental for high-quality pyrite phase formation. The justification for the modeling of hydrothermal iron pyrite is needed for this study, which could also broaden the impact on the growth of iron pyrite. Furthermore, the PH effect on the phase selection could be a good subject for the modeling efforts, therefore, a universal material predictive model will have broader impact (iron pyrite itself (or CaCO₃) may not be enough to guarantee this).

As the conclusion, the paper contains a significant amount of work done in modeling the PH effect on hydrothermally grown iron pyrite materials. It also contains the work to prove the PH effect in the previous experimental efforts. Having no universal predictive model on phase selection makes it an incomplete paper.

#####

We thank the reviewer for evaluating our manuscript, and address the concerns brought up below.

i. The justification for the modeling of hydrothermal iron pyrite is needed for this study, which could also broaden the impact on the growth of iron pyrite.

The goal of our work is indeed to establish a computationally-accessible approach for evaluating the synthetic accessibility of crystal polymorphs within hydrothermal growth methods. While it is true that FeS₂ pyrite has been widely studied as a potentially attractive optical absorber, our motivation in studying this material lies in its widely-documented but poorly understood polymorphism, which has been observed during hydrothermal growth. Thus, our goal is not to attempt to find the best synthesis method for optically active pyrite, but to understand how synthesis handles can lead to the growth of the pyrite versus marcasite phase in the first place. As our computational approach is

novel it is also important to pick a system for which there is good experimental data, further motivating the choice of FeS₂.

ii. Furthermore, the PH effect on the phase selection could be a good subject for the modeling efforts, therefore, a universal material predictive model will have broader impact (iron pyrite itself (or CaCO₃) may not be enough to guarantee this).

We believe that our analysis identifies the mechanism by which adsorption serves as a handle on phase selection, based on the example of FeS₂, through a general and transferrable quasi-thermodynamic approach. The derivation of our model provided in the supplementary methods relies on transition-state thermodynamics, as well as the general structure of the electrochemical double layer, potential-determining ions, ionic adsorption and ideal-solution thermodynamics. These concepts are certainly not limited to the hydrothermal growth of FeS₂, making our model readily transferrable to other mineral systems. Of course, we cannot claim that our work encompasses structure selection in every synthesis method – such would be an impossible challenge – but we do believe that the approach described provides a computationally-accessible metric of synthesizability by hydrothermal growth, as is the stated goal in the introduction. Such a metric is both novel and important in its ability to identify likely synthetically accessible metastable phases and the conditions under which they may form, which is an important step for computational materials prediction.

#####

Reviewer #3

The manuscript reports a model, based on ab initio thermodynamic methods, for the phase selection of pyrite vs marcasite phases of FeS₂ produced by hydrothermal methods. It is shown that the experimentally observed relationship between crystal phase and pH can be explained by a model which takes the surface energies of small FeS₂ crystals with adsorbed OH⁻ and H₃O⁺ species into account. The model correctly predicts the transition to within 2 pH units. Application of the model to other systems is discussed.

The problem considered is of genuine interest, and the computational methods chosen appear to be appropriate. The approximations made in the model appear to be reasonable. The paper is well written and clear for the reader (although it is a shame that so many of the details of the method are confined to the supplementary material). I believe that the originality of the model and the potential for the applicability of the methods developed here to a broader range of problems are sufficient to make it suitable for publication in Nature Communications.

#####

We thank the reviewer for evaluating our manuscript and appreciate the positive comments. We believe that the current structure of the manuscript is necessary to highlight our central message, which is that phase selection in the hydrothermal synthesis of FeS₂, and likely other mineral systems, may be evaluated in a quasi-thermodynamic sense, circumventing the difficulties of direct simulation of nucleation and growth. The methodological derivation and computational details, while important and, we believe, interesting in their own right, are lengthy and distract from this message. Thus, we feel that the current format, with the majority of the methodological detail confined to the supplementary information, is best suited for communicating our analysis.

Reviewers' Comments:

Reviewer #1 (Remarks to the Author):

I consider that the authors have modified the manuscript and supplementary information so that my questions and concerns are fully answered, and hence should be accepted as it.

Reviewer #2 (Remarks to the Author):

The authors have revised the submission accordingly.